# CircDOCK1 Regulates miR-186/DNMT3A to Promote Osteosarcoma Progression

**DOI:** 10.3390/biomedicines10123013

**Published:** 2022-11-23

**Authors:** Zhihui Jin, Jia Ye, Sen Chen, Yijun Ren, Weichun Guo

**Affiliations:** Department of Orthopedics, Renmin Hospital of Wuhan University, Wuhan 430060, China

**Keywords:** circDOCK1, miR-186, DNMT3A, OS, progression

## Abstract

Background: Circular RNAs (circRNAs), as a class of endogenous RNAs, are implicated in osteosarcoma (OS) progression. However, the functional properties of circDOCK1 in OS have been largely unexplored. The present study demonstrated the regulatory mechanism of circDOCK1 in OS. Methods: QRT-PCR and Western blots were used to determine the abundances of circDOCK1, miR-186, and DNMT3A. Cell counting kit-8 (CCK-8), 5-Ethynyl-2′-deoxyuridine (EdU), colony formation, Transwell, and wound healing assays were used to examine cellular multiplication, motility, and invasion. Luciferase reporter analysis, RNA immunoprecipitation (RIP), and pull-down assays were used to verify target relationships. Xenograft models were used to analyze in vivo function. Results: OS tissues and cells showed high levels of circDOCK1. By knocking down circDOCK1, cellular multiplication, motility, and invasion were suppressed. Furthermore, silencing circDOCK1 suppressed the growth of tumor xenografts. According to mechanistic studies, miR-186 targets DNA methyltransferases 3A (DNMT3A) directly and acts as a circDOCK1 target. Furthermore, circDOCK1 upregulated DNMT3A expression through sponging miR-186 to regulate the progression of OS. Conclusions: CircDOCK1 promotes OS progression by interacting with miR-186/DNMT3ADNMT3A, representing a novel therapeutic approach.

## 1. Introduction

Osteosarcoma (OS) is a primary osseous malignancy of mesenchymal cell origin with a high morbidity and mortality rate [1]. It is characterized by abnormal growth of bone-associated mesenchymal cells and is highly aggressive [2,3]. It was reported that about 20% of OS patients develop tumor metastases, including primarily pulmonary metastases and metastases toward lymph nodes or other soft tissues [4,5]. OS patients with tumor metastasis or relapse have a five-year survival rate of less than 30% [6,7]. Despite the clinical application of several therapeutic approaches, including surgery and neoadjuvant chemotherapy, their undesirable efficacy and serious adverse events require a deeper understanding of the mechanisms that regulate OS occurrence and progression in order to find new ways to treat this cancer.

Circular RNAs (circRNAs) are a class of noncoding RNAs that are produced by reversely splicing covalently linked 3′- and 5′-ends having no polyadenylate tails [8,9]. Due to their closed-loop architecture, circRNAs are more stable and virtually impervious to degradation by RNA exonucleases compared to their linear transcripts [10,11]. Numerous studies have found an association between differently expressed circRNAs and the incidence of OS and multiple other carcinomas. For example, hsa_circOMA1 promoted OS progression by modulating the mir-1294/c-Myc pathway [12]. Furthermore, a recent study found that circDOCK1 facilitates the cisplatin tolerance of OS via the miR-339–3p/IGF1R pathway [13]. In the GSE140256 dataset, has_circ_0020378 (circDOCK1) expression was upregulated in three OS tumor tissues relative to adjacent normal tissues. However, its role in the evolution of OS remains to be further explored.

From a functional perspective, circRNA can regulate the downstream mRNAs by serving as the competing endogenous RNAs (ceRNA) for microRNAs (miRNAs), resulting in various effects on cellular function [14,15,16]. There is growing evidence indicating the pivotal role of ceRNA networks in the carcinogenesis and evolution of OS. Thus, it is necessary to clarify the role of circRNA-mediated ceRNA networks in OS carcinogenesis and evolution.

## 2. Materials and Methods

### 2.1. Clinical Samples

Twenty-one pairs of OS tissues and healthy paracarcinoma controls were collected from the Renmin Hospital of Wuhan University. All patients provided written informed consent. The study protocol was approved by the Ethics Committee of the abovementioned university.

### 2.2. Bioinformatics Analysis

The GSE140256 dataset was used to screen the differently expressed circRNAs. The miR-186–circDOCK1 binding sites were predicted using the Circinteractome database. The potential targets of miR-186 were identified by the TargetScan database.

### 2.3. Cell Culture and Transfection

We cultured the normal hFOB 1.19 osteoblasts and OS (HOS and U2OS) cells in DMEM (Gibco, USA) involving FBS (10%; Biological Industries, Brisbane, Australia) and penicillin/streptomycin (1%) in a 5% CO_2_ and 37 °C incubator. DNMT3A overexpression and pcDNA3.1 vectors were synthesized by Servicebio (Wuhan, China). Short heparin RNA (shRNA) for circDOCK1 (sh-circDOCK1) and shRNA negative control (NC) were obtained from Sangon Biotech (Shanghai, China). GenePharma (Shanghai, China) synthesized miR-186 inhibitors, mimics, and NC inhibitors or NC. Transfection was carried out using Lipofectamine 3000.

### 2.4. RNA Extraction and qRT-PCR

Total RNA was extracted from tissue and cells using TRIzol reagent (Sigma, St. Louis, MO, USA). The cDNA was synthesized via a kit from Takara (Shanghai, China). The circRNA, mRNA, and miRNA were amplified using the SYBR Green PCR kit, also from Takara (Shanghai, China). Appendix A represents the primer information. Relative expressions were computed and evaluated through the 2^–ΔΔCt^ approach. A GAPDH internal control was used for circRNA and mRNA, whereas a U6 internal control was used for miRNA. The primers were listed in Appendix A.

### 2.5. Ribonuclease R (RNase R) Digestion

A total of 20 U/µL RNase R was incubated with 3 ug RNA for 15 min at 37 °C. In the following step, qRT-PCR was used to detect the expression of circDOCK1 and liner DOCK1.

### 2.6. Sanger Sequencing

The amplification products of circRNA in a T vector were used for Sanger sequencing. The primers were synthesized and designed to verify the back splice junction of circDOCK1.

### 2.7. Fluorescence In Situ Hybridization (FISH)

For the FISH assay, Cy3-labelled circDOCK1 probes (CTCAAGGAAAGGTAGTCTTAACA) and FAM-labelled miR-186 probes (GCCAACCTCACAAGACAACAAT) were used [13]. Briefly, the cells were immobilized in paraformaldehyde (4%) and hybridized overnight in a humidified incubator at 37 °C. After PBS washing, the cells were counterstained with DAPI. Finally, images were captured using a fluorescent microscope (Zeiss, Jena, Germany).

### 2.8. Cell Proliferation

The Cell Counting Kit-8 (Beyotime, Shanghai, China) was used to assess the proliferation potential of OS cells. The cells were seeded into a 96-well plate added with CCK-8 reagent at 10 µL/well. Following 1 h of incubation, a 450-nm optical density was observed with a microplate reader. In addition, the EdU cell proliferation kit (Beyotime, Shanghai China) was used to complete the EdU assay. Following incubation in EdU solution, the OS cells were immobilized in paraformaldehyde (4%) and stained with click additive solution. After that, a fluorescent microscope (Zeiss, Jena, Germany) was used for image capturing.

### 2.9. Colony Formation Assay

In 6-well microplates, transfected cells were cultured for 10 days. Afterward, cells were immobilized in paraformaldehyde (4%) and stained using crystal violet (0.1%) before counting.

### 2.10. Transwell Analysis

Cell invasion was observed through the Transwell assessment. An upper chamber precoated with Matrigel was added to the OS cells in a serum-free medium, while a lower chamber was added to a 10% serum-involving medium (600 µL). After 24 h of incubation, cells were immobilized and stained with crystal violet (0.1%). The invasive cells were photographed microscopically (Zeiss, Jena, Germany).

### 2.11. Wound-Healing Assay

A 6-well microplate was inoculated with cells until an 80% confluency was reached. A vertical line was drawn in the cellular layer with a 10 µL pipette. Then, the cells were PBS-washed and incubated for 48 h. Images at 0 and 48 h were captured with a microscope, and wound width was estimated using ImageJ software.

### 2.12. Dual-Luciferase Reporter Analysis

To construct the luciferase reporter vector, we cloned the DNMT3A or circDOCK1 sequences with the mutant-type (mut) or wild-type (wt) miR-186 binding sites into the pGL3 vector. Luciferase reporter vectors and NC or miR-186 mimics were co-transfected into OS cells using Lipofectamine 2000. The subsequent step was the reporter assay system (dual luciferase, Promega, Madison, WI, USA)-based assessment of luciferase activity.

### 2.13. RNA Immunoprecipitation (RIP) Assay

A Magna RIP kit (Sigma, St. Louis, MI, USA) was used to perform the RIP analysis. After the initial lysis of the OS cells, the lysate was incubated overnight with anti-Ago2 or anti-IgG antibody-coated beads. Enrichment of DNMT3A, circDOCK1, or miR-186 was studied via qRT-PCR.

### 2.14. Western Blotting

RIPA buffer was used to separate the total cellular protein, and then the BCA assay kit (Beyotime) was used to quantify the protein level. Following the electrotransfer of the protein onto the PVDF membranes, the membranes were incubated overnight using primary anti-DNMT3A (Proteintech, Wuhan, China) and anti-β-Actin (Servicebio, Wuhan, China) antibodies and subsequent 2-h incubation with corresponding HRP-labeled secondary antibody IgG. Finally, X-ray films were used to detect the emitted light.

### 2.15. Xenograft Model

We used sh-circDOCK1 or sh-NC to stably transfect the U2OS cells, followed by subcutaneous injection of these transfected cells into 6 nude mice (n = 3 per group). Tumor growth and volume were supervised every five days, 6 times. The mice were euthanized 30 days later for tumor samples, which were then weighted at the endpoint. The expression of circDOCK1, miR-186, and DNMT3A in the tumors was observed using qRT-PCR or immunohistochemistry.

### 2.16. Statistical Analysis

The results are presented as means ± SD. Pearson’s correlation coefficient was used to determine the correlation between circDOCK1, miR-186, and DNMT3A. The difference was evaluated using Student’s *t*-test (pairwise comparisons) or analysis of variance (comparing multiple groups). *p*-values less than 0.05 were regarded as statistically significant.

## 3. Results

### 3.1. CircDOCK1 Expression Is Highly Expressed in OS Tissues and Cells

According to the GSE140256 dataset, the top three upregulated circRNAs (hsa_circ_0010220, hsa_circ_0000253 and hsa_circ_0020378) were noted in OS tissues compared to in healthy paracarcinoma tissues (Figure 1A). Then, qRT-PCR results showed that hsa_circ_0020378 (circDOCK1) was the most upregulated circRNA in OS cells compared to hFOB cells (Figure 1B). Thus, circDOCK1 was chosen for further study. Following this, further validation of the circular structure of circDOCK1 in OS was performed. Sanger sequencing confirmed the back splice junction of circDOCK1 (Figure 1C). Afterward, agarose gel electrophoresis showed the presence of circDOCK1 only in the divergent primer-amplified cDNA, which was absent in gDNA (Figure 1D). Furthermore, circDOCK1 was found to be more resistant to RNase R treatment than linear DOCK1 (Figure 1E). Moreover, OS tissues and cells expressed high levels of circDOCK1 (Figure 1F,G). The above findings suggested the stability and upregulation of circDOCK1 in OS.

### 3.2. CircDOCK1 Knockdown Inhibits OS Progression In Vitro

To explore how circDOCK1 affects OS progression in vitro, the junction site of circDOCK1 was targeted by two shRNAs (sh-circDOCK1#1, sh-circDOCK#2). Figure 2A shows that two shRNAs significantly suppressed circDOCK1 expression. Since sh-circDOCK1#1 had higher knockdown efficiency, it was used for further experiments. According to CCK-8 and EdU assays, silencing circDOCK1 significantly decreased cell proliferation (Figure 2B,C). Furthermore, circDOCK1 knockdown also decreased colony formation (Figure 2D). Moreover, circDOCK1 knockdown decreased the invasive potential and motility of OS cells (Figure 2E,F). According to these results, it is found that silencing circDOCK1 inhibited OS progression in vitro.

### 3.3. CircDOCK1 Acts as a Sponge for miR-186

To investigate the mechanism of how circDOCK1 regulates OS progression, Circinteractome, Starbase and circBank were used to predict the circDOCK1 targets, which indicated the possibility of miR-186 and miR-339-3p as circDOCK1 targets (Figure 3A). Then, qRT-PCR results showed that miR-186 was the most downregulated miRNA in OS cells compared to hFOB cells (Figure 3B). Thus, we chose miR-186 for further study. The circDOCK1 reporter was used to validate the miRNAs that could bind to circDOCK1. The luciferase activity of the circDOCK1 reporter was significantly decreased after transfection of sh-circDOCK1 (Figure 3C). Bioinformatics analysis was used to predict the circDOCK1–miR-186 binding sites (Figure 3D). Based on subsequent dual luciferase reporter analysis, the miR-186 mimics led to an apparent weakening of luciferase activity in the circDOCK1-wt group, while it did not affect the circDOCK1-mut group (Figure 3E). Furthermore, the RIP assay found enrichment of circDOCK1 and miR-186 by anti-Ago2 but not by anti-IgG (Figure 3F). In addition, the FISH assay revealed the cytoplasmic co-localization of circDOCK1 and miR-186 (Figure 3G). A drastic decline in miR-186 expression was also observed in OS tissues and cells (Figure 3H,I). Additionally, miR-186 expression was negatively regulated by circDOCK1 (Figure 3J). Moreover, circDOCK1 knockdown increased miR-186 expression (Figure 3K). These findings exposed the miR-186-sponging role of circDOCK1.

### 3.4. Knockdown of miR-186 Partly Reverses the Effect of sh-circDOCK1 on OS Progression

We used sh-circDOCK1 and miR-186 inhibitors to co-transfect the OS cells in order to explore the association of circDOCK1 with miR-186. MiR-186 expression was elevated after the circDOCK1 knockdown, which, however, dropped after transfecting the OS cells with miR-186 inhibitors (Figure 4A). Furthermore, miR-186 knockdown overturned the effects of circDOCK1 knockdown on cell proliferation (Figure 4B,C,F). Moreover, mir-186 downregulation attenuated the inhibition of cellular motility and invasion mediated by circDOCK1 knockdown (Figure 4D,E,G,H). These results showed that miR-186 knockdown partly reverses the effect of sh-circDOCK1 on OS progression.

### 3.5. DNMT3A Was a Target of miR-186

The potential targets of miR-186 were predicted using bioinformatics analysis. Three genes (DNMT3A, MYLIP, FBXL5) were recognized as the possible targets of miR-186 (Figure 5A). Subsequently, qRT-PCR revealed that the miR-186 mimics suppressed the expressions of three target genes, particularly the expression of DNMT3A (Figure 5B). Thus, it was selected for subsequent experiments. Estimating miR-186–DNMT3A binding sites was conducted by bioinformatics means (Figure 5C). Based on subsequent dual luciferase reporter analysis, the miR-186 mimics attenuated the luciferase activity in the DNMT3A-wt group, whereas it rarely affected the DNMT3A-mut group (Figure 5D). Furthermore, high DNMT3A expressions were found in the OS tissues (Figure 5E). Additionally, miR-186 significantly decreased the DNMT3A protein level (Figure 5F). Importantly, DNMT3A expression was negatively correlated with miR-186 expression (Figure 5G).

Following this, we further explored the relationships among circDOCK1, miR-186, and DNMT3A. DNMT3A level was found to be positively associated with circDOCK1 (Figure 5H). Based on qRT-PCR and Western blotting assays, circDOCK1 knockdown was found to inhibit the mRNA and protein expressions of DNMT3A, which were rescued by miR-186 inhibitors in OS cells (Figure 5I,J). These results indicated that DNMT3A was a target of miR-186.

### 3.6. CircDOCK1 Regulates OS Progression through the miR-186/DNMT3A Axis

To investigate how the circDOCK1/miR-186/DNMT3A axis functions in OS evolution, we transfected OS cells with sh-circDOCK1 and DNMT3A overexpression vectors. Based on CCK-8, EdU, Transwell invasion, and scratch-wound assays, the miR-186 inhibitors and DNMT3A restored the sh-circDOCK1-elicited effects on cellular multiplication, motility, and invasion (Figure 6A–G).

### 3.7. CircDOCK1 Knockdown Suppressed OS Growth In Vivo

A xenograft model was used to validate the in vivo function of circDOCK1. It was found that the mass and volume of tumors in the sh-circUSP34 group were smaller than in the sh-NC group (Figure 7A–C). QRT-PCR results showed that miR-186 expression was significantly enhanced, whereas circDOCK1 and DNMT3A expressions were reduced in the sh-circUSP34 group (Figure 7D–F). Similarly, the immunofluorescence assay revealed that DNMT3A protein levels weres reduced in the sh-circUSP34 group (Figure 7G). Furthermore, IHC staining demonstrated that Ki-67 protein expression increased after the knockdown of circDOCK1 (Figure 7H). These results indicated that circDOCK1 knockdown suppressed OS growth in vivo.

## 4. Discussion

In young adults and children, OS is a common osseous malignancy [17,18]. Although the five-year survival was elevated over the last 3 decades for the OS population, their prognosis is still poor, making it necessary to discover new diagnostic and therapeutic targets for the management of OS [19,20,21]. The advantages of circRNA, such as its structural stability and tissue specificity, have attracted the attention of researchers in recent years [22,23]. Recent evidence suggests the close association of circRNA dysregulation with carcinoma progression and metastasis, including OS, indicating its potential as an OS therapeutic target [24,25,26,27].

Several circRNAs have been reported to play an oncogenic role in OS progression through malignant phenotype promotion, including circOMA1, circUSP34, and circNT5C2 [12,28,29]. In this study, circDOCK1, a novel circRNA, was chosen by screening the GSE140256 dataset. Differently expressed circRNAs between OS and adjacent normal tissues were identified through a microarray profiling analysis. The present study confirmed the upregulation of circDOCK1 in the OS tumor samples and cells. Further loss-of-function assays showed that knocking down circDOCK1 inhibited OS cell multiplication, motility, and invasion. Additionally, we determined that circDOCK1 knockdown suppresses tumor growth in vivo by creating a nude mouse model of OS xenografts. As implied by the foregoing findings, circDOCK1 plays an OS-promoting role, which possibly serves as a therapeutic target for OS.

In OS, the circRNA/miRNA/mRNA axis represents a common mechanism underpinning circRNA’s functionality [30,31,32]. Growing evidence confirms the regulatory action of miRNAs on the progression of diverse carcinomas, which can either facilitate or prevent malignancies [33]. The present study found that circDOCK1 negatively regulates miR-186, one of its targets. Numerous studies have suggested that miR-186 may suppress diverse tumor types, such as papillary thyroid, esophageal and gastric carcinomas [34,35,36]. According to Tan et al., circ0038632 promoted the malignant phenotype of OS cells through the miR-186/DNMT3A axis [37]. Additionally, Xiao et al. implied the correlation of mir-186 downregulation with the metastasis of tumors. Previous reports have indicated the anti-OS role of miR-186 [38]. Similarly, this study confirmed the OS-inhibitory function of miR-186 [39,40]. The miR-186 knockdown evidently facilitated malignant OS cell phenotype. Furthermore, the present study elucidated that the regulation of OS progression by circDOCK1 was achieved via miR-186 sponging.

It is worth noting that circRNAs should only block the expressions of targeted mRNA but not the level of miRNA when they act as miRNA sponges. However, we found that silencing circDOCK1 increased miR-186 expression in OS cells. This phenomenon might be associated with the following reasons. In addition to acting as miRNA sponges, some circRNAs could encode proteins in a cap-independent manner [8]. These proteins may function as transcriptional repressors or activators to control pri-miRNA transcription, thus affecting the expression of miRNAs [41]. Based on the previous reports, we speculate that circDOCK1 may encode a protein that functions as a transcriptional repressor to inhibit the expression of miR-186. Nevertheless, this hypothesis still needs to be further verified by in-depth research.

After further investigation, DNMT3A was verified as a mir-186 target in OS. DNMT3A is an epigenetic enzyme engaged in cell differentiation, embryonic development and epithelial-to-mesenchymal transition [42]. The enhanced catalytic activity of DNMT3A is associated with cancer development and progression. Cheng et al. reported that DNMT3A facilitated OS progression by mediating miR-149 DNA methylation to activate the Notch1/Hedgehog pathway [43]. In line with the former study, DNMT3A’s carcinogenic role in OS was also recognized herein, and miR-186 suppressed OS by decreasing DNMT3A. Furthermore, it was confirmed that by acting as a mir-186 sponge, circDOCK1 causes DNMT3A upregulation to mediate the progression of OS.

However, there are some limitations in our research. Three mice per group is not a robust sample size. Thus, there is a need for more prospective clinical studies and larger sample sizes to evaluate the robustness of the proposed therapeutic approach and its feasibility in clinical practice.

## 5. Conclusions

In conclusion, circDOCK1 regulated OS progression both in vitro and in vivo through miR-186/DNMT3A axis modulation and elevated expression in OS. Furthermore, the present study provides a novel therapeutic target for OS in addition to providing a new understanding of its pathology.

## Figures and Tables

**Figure 1 biomedicines-10-03013-f001:**
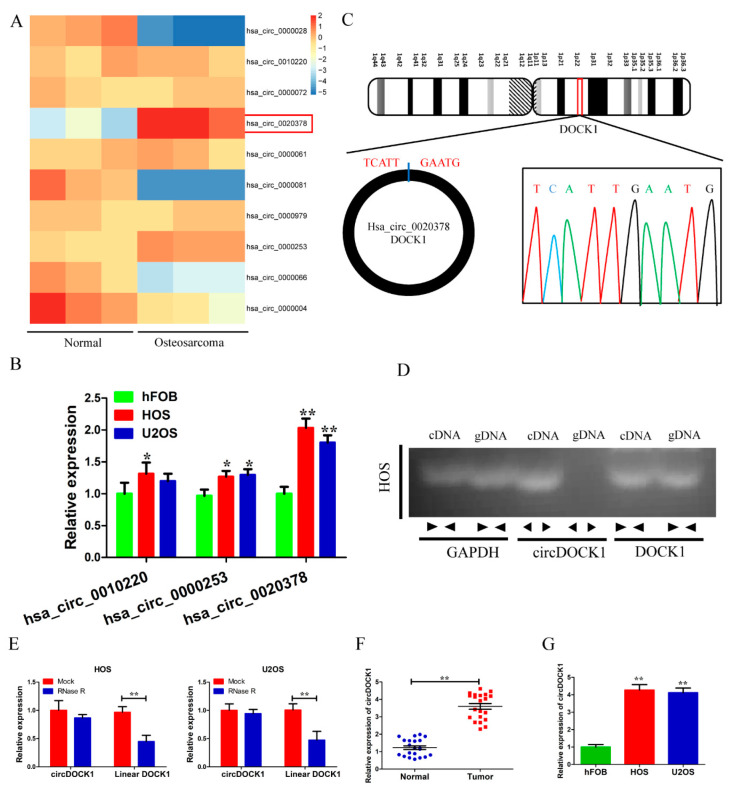
CircDOCK1 is highly expressed in OS tissues and cells. (**A**) CircRNAs are expressed differently between OS and healthy paracarcinoma tissues. (**B**) The expressions of top three upregulated circRNAs in OS cell lines. (**C**) Sanger sequencing validated the back splice junction of circDOCK1. (**D**) PCR products of circDOCK1 in agarose gel electrophoresis. (**E**) CircDOCK1 was stable after RNase R treatment. (**F**) CircDOCK1 expression in OS samples. (**G**) The expression of circDOCK1 in OS cell lines. * *p* < 0.05, ** *p* < 0.01.

**Figure 2 biomedicines-10-03013-f002:**
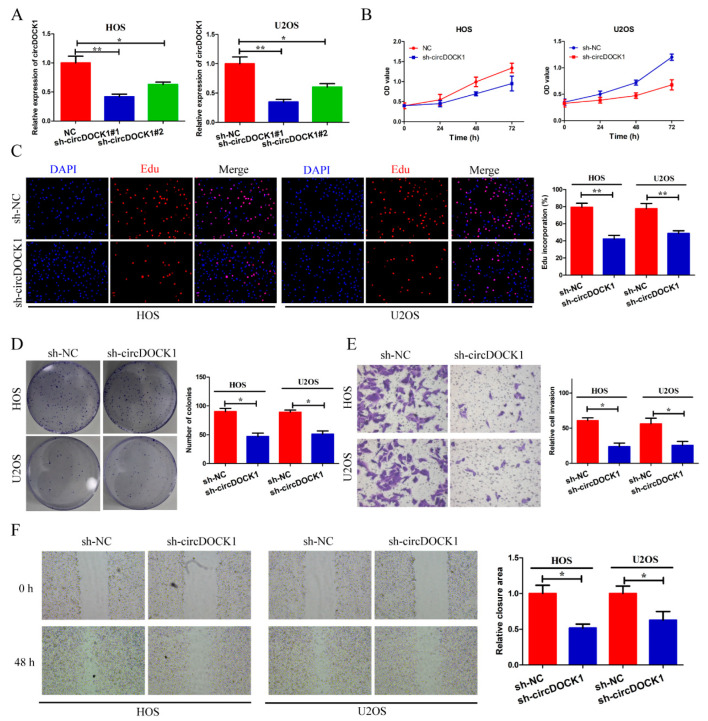
CircDOCK1 knockdown inhibits OS progression in vitro. (**A**) CircDOCK1 expression was detected in transfected OS cells. (**B**,**C**) CCK-8 plus EdU assay outcomes for cellular proliferative potential. (**D**) Colony formation efficiency was detected in transfected OS cells. (**E**,**F**) Wound-healing and Transwell assay outcomes for cellular motility and invasion. * *p* < 0.05, ** *p* < 0.01.

**Figure 3 biomedicines-10-03013-f003:**
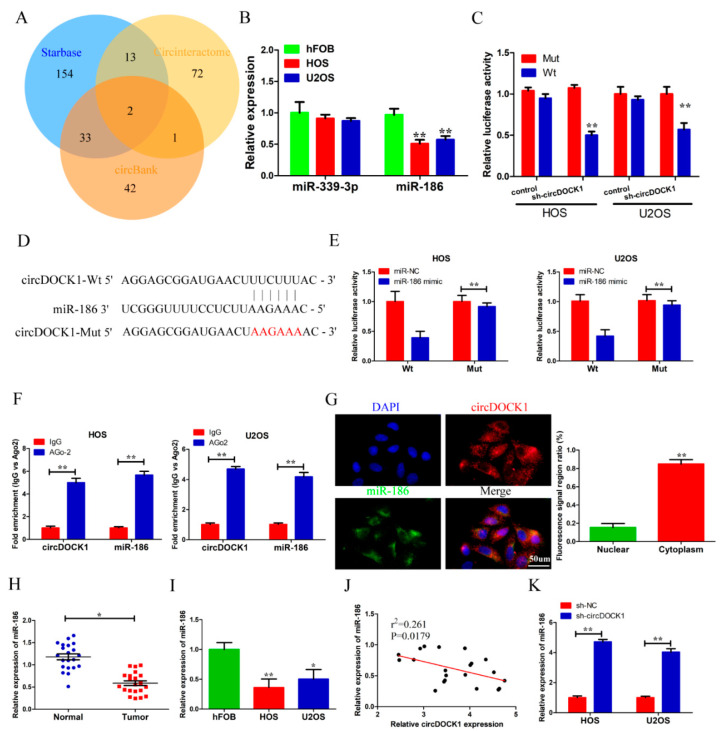
CircDOCK1 acts as a sponge for miR-186. (**A**) Bioinformatics analysis outcomes predicting the circDOCK1 targets. (**B**) The expressions of two target miRNAs in OS cell lines. (**C**) Luciferase activity of the circDOCK1 reporter after transfection of sh-circDOCK1 in OS cells. (**D**) The circDOCK1–miR-186 binding sites were predicted using bioinformatics, and the red sequence means mutant binding sequences. (**E**) Luciferase activity outcome on co-transfected OS cells. (**F**) RIP assay outcome on the binding correlation of circDOCK1 with miR-186. (**G**) FISH assay outcome suggests the cytoplasmic co-localization of circDOCK1 and miR-186. (**H**,**I**) The OS tissue and cellular levels of MiR-186 were markedly decreased. (**J**) The miR-186 level was negatively regulated by circDOCK1. (**K**) Silencing circDOCK1 increased miR-186 expression. * *p* < 0.05, ** *p* < 0.01.

**Figure 4 biomedicines-10-03013-f004:**
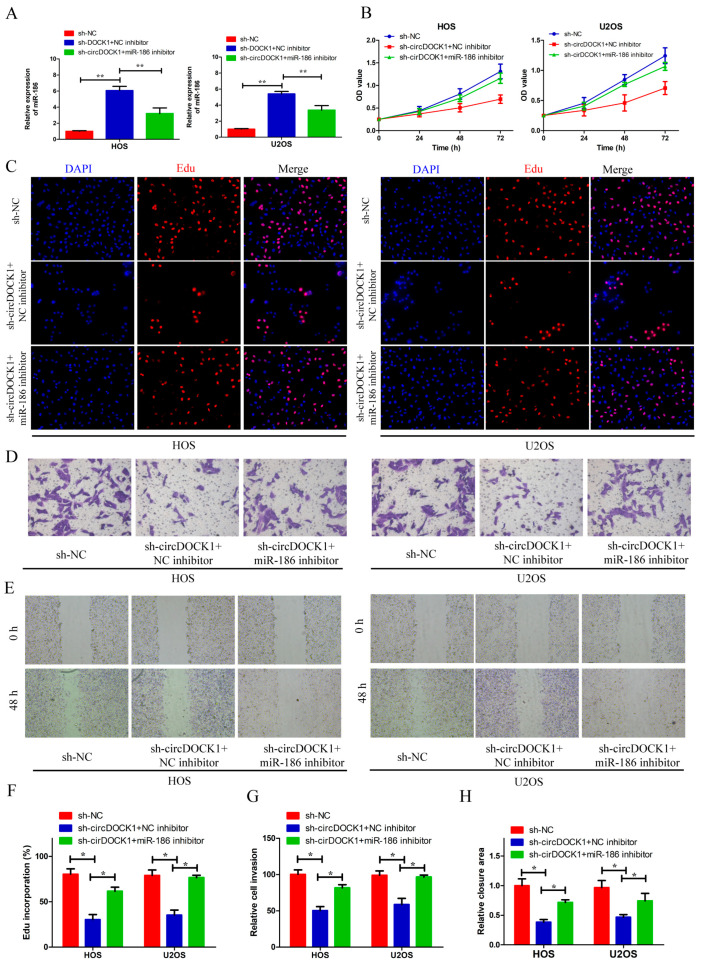
MiR-186 knockdown partly reverses the effect of sh-circDOCK1 on OS progression. (**A**) The miR-186 level was evaluated in OS cells following transfection with inhibitors of miR-186 and sh-circDOCK1. (**B**,**C**,**F**) CCK-8 and EdU assay outcomes concerning cellular proliferative potential. (**D**,**E**,**G**,**H**) Wound-healing and Transwell assay findings for cellular motility and invasion. * *p* < 0.05, ** *p* < 0.01.

**Figure 5 biomedicines-10-03013-f005:**
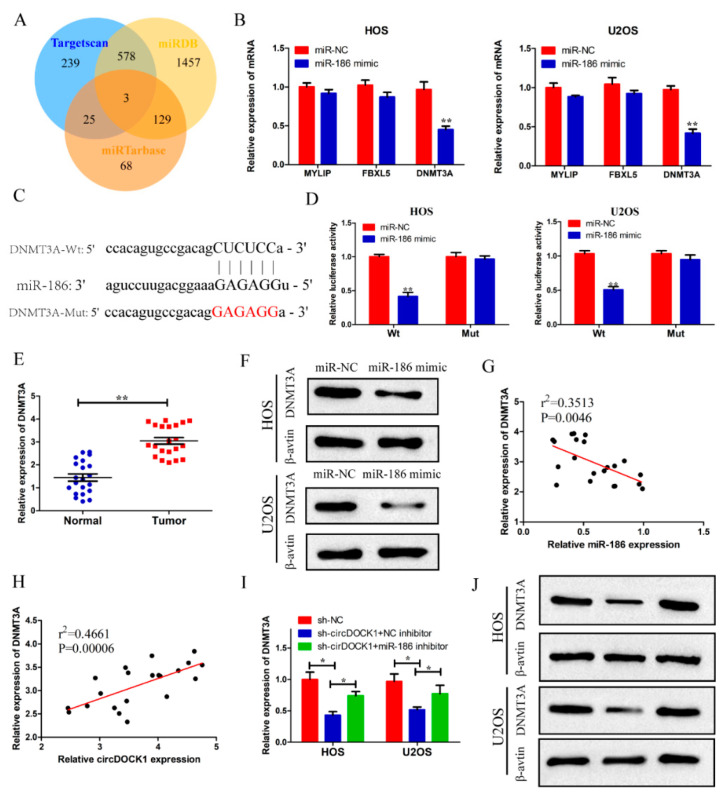
CircDOCK1 upregulated DNMT3A expression via sponging miR-186. (**A**) Bioinformatics analysis outcomes predicting the potential miR-186 target genes. (**B**) The expressions of four target genes were reduced after transfection with miR-186 mimics. (**C**) The miR-186–DNMT3A binding sites were predicted using bioinformatics, and the red sequence means mutant binding sequences. (**D**) Luciferase reporter assay-based validation concerning the correlation of DNMT3A with miR-186. (**E**) DNMT3A expression was high in OS tissues. (**F**) MiR-186 significantly decreased DNMT3A protein levels. (**G**) Expression of DNMT3A was negatively associated with miR-186. (**H**) DNMT3A level was positively associated with circDOCK1. (**I**,**J**) After co-transfection, qRT-PCR combined with Western blotting was employed to assay the protein and mRNA levels of DNMT3A. * *p* < 0.05, ** *p* < 0.01.

**Figure 6 biomedicines-10-03013-f006:**
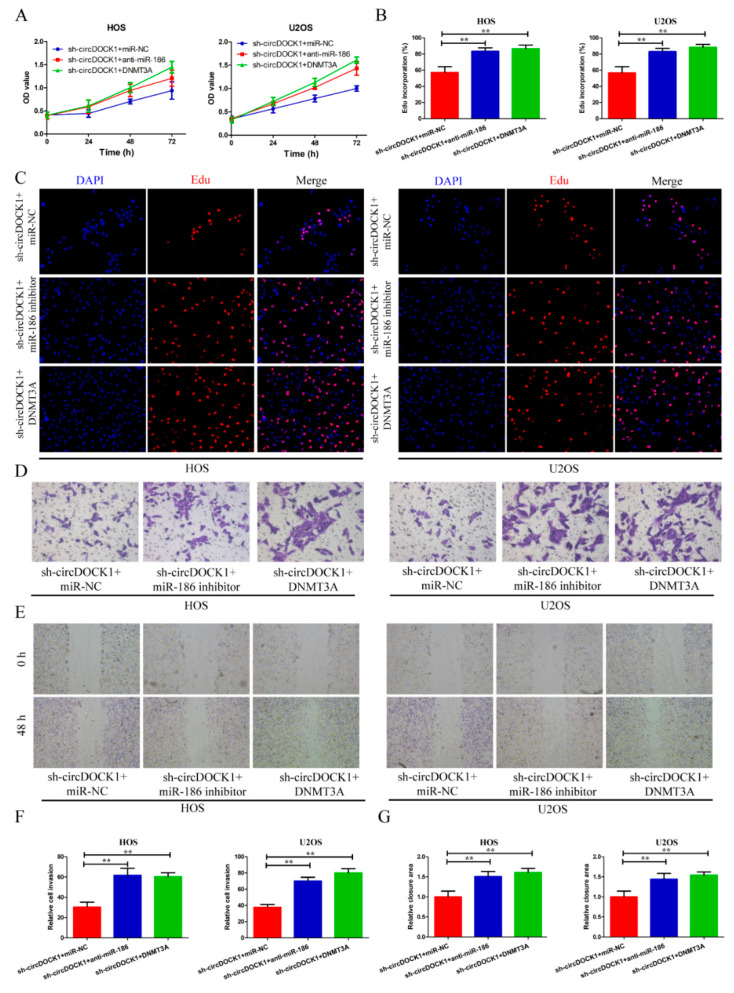
CircDOCK1 regulates OS progression through the miR-186/DNMT3A axis. (**A**–**C**) After co-transfection, the cellular proliferative potential was examined via CCK-8 and EdU assays. (**D**–**G**) Wound-healing and Transwell assay outcomes for cellular motility and invasion. ** *p* < 0.01.

**Figure 7 biomedicines-10-03013-f007:**
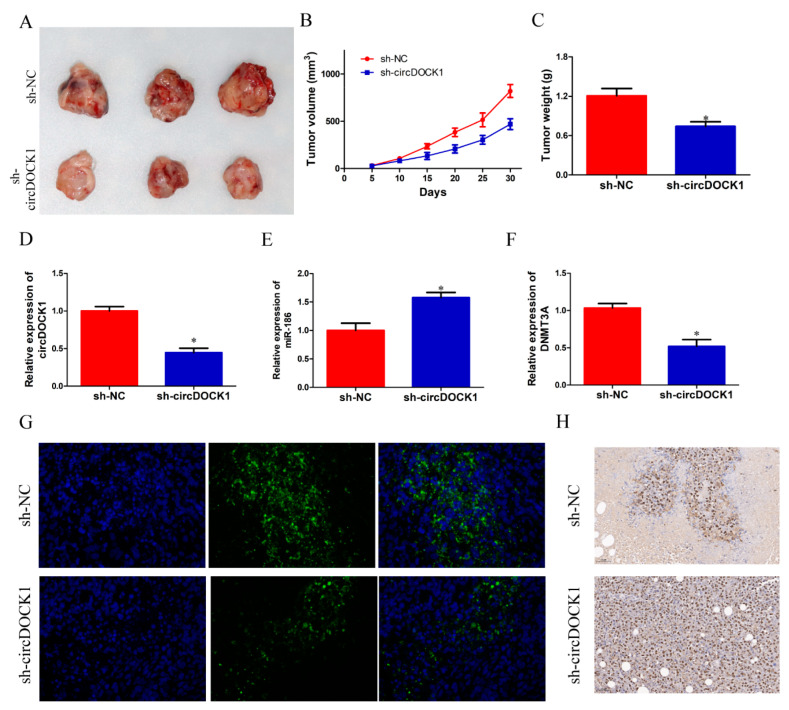
CircDOCK1 knockdown suppressed OS growth in vivo. (**A**) The tumor mass images. (**B**,**C**) The volume and weight information of tumors in both groups. (**D**–**F**) The tumor tissue levels of circDOCK1, DNMT3A, and miR-186. (**G**) Immunofluorescence staining of DNMT3A. (**H**) Ki-67 staining of tumor tissues. * *p* < 0.05.

## Data Availability

Datasets are available upon request.

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
