# Peer review of "CircDOCK1 Regulates miR-186/DNMT3A to Promote Osteosarcoma Progression"

_biomedicines, 2022, doi:10.3390/biomedicines10123013_

Round 1

Reviewer 1 Report

Authors present a paper where they intend to exploit the potential of circDOCK1 in the regulation of OS growth potential.

The experimental plan is appropriate and multidisciplinary, and the results give more than satisfactory support to the role of the upregulation of circDOCK1 in OS proliferation, motility, and invasion potential. Noteworthy is the demonstration that circDOCK1 knockdown suppresses tumor growth in vivo and that the putative mechanism is via miR-186 sponging and downregulation of DNMT3A, thus confirming an emerging role for circDOCK1 as therapeutic target for OS and, possibly, for other solid highly metastatic tumors. I would have compiled a more complete and exhaustive Discussion to fully explain the real potential of this work, although it is sufficient in the present form.

These significant data, in my opinion, endorse this work for publication in this Journal.

Author Response

These significant data, in my opinion, endorse this work for publication in this Journal.

Response: We are incredibly grateful for your comments. We will continue to complete the relevant study with high standards and strict requirements

Reviewer 2 Report

The authors of this manuscript investigated the regualtory mechanism of circDOCK1 in osteosarcoma (OS) development. They assessed its increased expression in 21 OS samples compared to related healthy tissue, and showed that its silencing with shRNAs decreased cell proliferation, motility and invasion potential in OS cell lines. Moreover, they reported the colocalization and functional interaction of circDOCK1 with miR-186, as well as upregulation of DNMT3A. They also performed an in vivo validation taking advantage of a xenograft murine model in which they confirmed that circDOCK1 silencing suppress OS growth.

The manuscript is well written and data are clearly presented. The topic is of interest for researchers in the field.

However, some issues need to be addressed:

1) Role of miRNA and epigenetics in OS should be properly discussed, especially given the role of DNMT3A in methylation. Several evidences exixst on this topic as in the following reference:

- Hattinger CM, Patrizio MP, Tavanti E, et al. Genetic testing for high-grade osteosarcoma: a guide for future tailored treatments?. Expert Rev Mol Diagn. 2018;18(11):947-961. doi:10.1080/14737159.2018.1535903

2) Several similar studies focusing on the role of circular RNAs in OS in the same axis have been recently published and should be cited:

- Tan X, Zeng C, Li H, Tan Y and Zhu H (2022) Circ0038632 modulates MiR-186/DNMT3A axis to promote proliferation and metastasis in osteosarcoma. Front. Oncol. 12:939994. doi: 10.3389/fonc.2022.939994

3) Limitations of the study should be clearly mentioned, e.g  the need of more prospective clinical studies and larger sample sizes (3 mice per group is not really a robust sample size) to evaluate the robustness of the proposed therapeutic approach and its feasibility in clinical practice.

Author Response

1) Role of miRNA and epigenetics in OS should be properly discussed, especially given the role of DNMT3A in methylation. Several evidences exixst on this topic as in the following reference: Hattinger CM, Patrizio MP, Tavanti E, et al. Genetic testing for high-grade osteosarcoma: a guide for future tailored treatments?. Expert Rev Mol Diagn. 2018;18(11):947-961. doi:10.1080/14737159.2018.1535903

Response: Thanks for your critical comments. Based on your comments, we have properly discussed the role of miRNA and epigenetics in OS. Besides, the article (Hattinger CM, Patrizio MP, Tavanti E, et al. Genetic testing for high-grade osteosarcoma: a guide for future tailored treatments? Expert Rev Mol Diagn. 2018;18(11):947-961. doi:10.1080/14737159.2018.1535903) was cited in our revised manuscript (reference 18)

2) Several similar studies focusing on the role of circular RNAs in OS in the same axis have been recently published and should be cited: Tan X, Zeng C, Li H, Tan Y and Zhu H (2022) Circ0038632 modulates MiR-186/DNMT3A axis to promote proliferation and metastasis in osteosarcoma. Front. Oncol. 12:939994. doi: 10.3389/fonc.2022.939994

Response: Thanks for your critical comments. Based on your suggestion, we have cited the article in our revised manuscript (reference 37)

3) Limitations of the study should be clearly mentioned, e.g the need of more prospective clinical studies and larger sample sizes (3 mice per group is not really a robust sample size) to evaluate the robustness of the proposed therapeutic approach and its feasibility in clinical practice.

Response: Thanks for your critical comments. We have clearly mentioned the limitations of study in the discussion section. The limitations mentioned above were all included in our revised manuscript.

Reviewer 3 Report

The manuscript describes a new axis of regulation by circDOCK where it is shown to song miR186 that in turn regulate DNMT3A. The authors did a very thorough analysis to establish the clinical relevance of circDOCK1, miR186 in OS. However, the data presented does not provide a convincing evidence of the regulatory axis.

Below are other major concerns:

1. The manuscript does not expand on why they choose circDOCK1 Even more missing connection is  why miR186 was selected. They cite a paper where circDOCK1 is shown to sponge another miRNA (miR339).  CircInteractome must have identified more miRNAs targets for this circRNA. They should show a bioinformatic analysis just like in figure 5a to show the identification of miR186.

2. The method section should be improved. It is not clear what probes were used for FISH analysis. There are no references. The figure for FISH is not very convincing. there is no scale bar and there is no quantification of the extent of co localization seen.

3. The PCR validation in figure 1 should also include linear DOCK1 and circDOCK1. It is confusing as it stand right now

4. The authors keep using circUSP48 in many figures without explaining what it is?

The data in figure 5 and its conclusions does not establish axis. The luciferase assay should be done in circDOCK1 KD cells and that should show the expected change.. right now, it only suggest that miR186 has an effect on DNMT3 levels.

5. Also, an overwhelming issue is that loss of miR186 expression in circDOCK1 KD doesnot make sense when circRNA is acting as a sponge, it should only block the function ( level of DNMT3A) but not the level of miRNA. Authors should provide a discussion of these findings in the manuscript.

Author Response

1. The manuscript does not expand on why they choose circDOCK1 Even more missing connection is why miR186 was selected. They cite a paper where circDOCK1 is shown to sponge another miRNA (miR339). CircInteractome must have identified more miRNAs targets for this circRNA. They should show a bioinformatic analysis just like in figure 5a to show the identification of miR186.

Response: Thanks for your critical comments. According to GSE140256 dataset, the top three upregulated circRNAs (hsa_circ_0010220, hsa_circ_0000253 and hsa_circ_0020378) were screened in OS tissues than in adjacent normal tissues. Then, qRT-PCR results showed that hsa_circ_0020378 (circDOCK1) was the most upregulated circRNA in OS cells compared to hFOB cell. Thus, circDOCK1 was chosen for further study. In order to clearly present information to the reader, the reason why we choose circDOCK1 was added in our revised manuscript and the qRT-PCR results were added in Figure 1.

Besides, the identification of miR-186 was also added in our revised manuscript. Circinteractome, Starbase and circBank were used to predict the circDOCK1 targets, which indicated the possibility of miR-186 and miR-339-3p as circDOCK1 targets. Then, qRT-PCR results showed that miR-186 was the most downregulated miRNA in OS cells compared to hFOB cell. Thus, we choose miR-186 for further study. The above results were added in Figure 3 and the revised manuscript.

2. The method section should be improved. It is not clear what probes were used for FISH analysis. There are no references. The figure for FISH is not very convincing. there is no scale bar and there is no quantification of the extent of co localization seen.

Response: Thanks for your critical comments. Based on your suggestion, we have added the probe sequences and relevant references in method section. Besides, the scale bar and quantification of the extent of co-localization were added in Figure 3.

3. The PCR validation in figure 1 should also include linear DOCK1 and circDOCK1. It is confusing as it stand right now

Response: Thanks for your critical comments. Based on your comments, the linear DOCK1 and circDOCK1 was included in PCR validation, and the figure has been changed in Figure 1.

4. The authors keep using circUSP48 in many figures without explaining what it is? The data in figure 5 and its conclusions does not establish axis. The luciferase assay should be done in circDOCK1 KD cells and that should show the expected change. right now, it only suggest that miR186 has an effect on DNMT3 levels.

Response: We are very sorry for these mistakes in our figures. Actually, circUSP48 was another research direction of our team. When we saved the pictures, the names of the image were saved as the previous name by default. In order to solve this problem, we have carefully checked all the mistakes in our figures and corrected all the wrong names.

In addition, we admitted that the data in figure 5 and its conclusions does not establish axis. As you have pointed out that it only suggests that miR186 has an effect on DNMT3A levels. Based on your suggestion, the conclusion of figure 5 has changed to that DNMT3A was a target of miR-186. Besides, the luciferase assay was done in circDOCK1 KD cells and the results were added in Figure 3.

5. Also, an overwhelming issue is that loss of miR186 expression in circDOCK1 KD doesnot make sense when circRNA is acting as a sponge, it should only block the function (level of DNMT3A) but not the level of miRNA. Authors should provide a discussion of these findings in the manuscript.

Response: Thanks for your critical comments. We agree with your point of view. By reviewing the literature, we think this phenomenon may be associated with the following reasons. In addition to acting as miRNA sponges, some circRNAs could encode proteins in a cap-independent manner. These proteins may function as transcriptional repressors or activators to control pri-miRNA transcription, thus affecting the expression of miRNAs. Based on the previous reports, we speculate that circDOCK1 may encode a protein which function as transcriptional repressors to inhibit the expression of miR-186. Nevertheless, this hypothesis still needs to be further verified by in-depth research. Besides, we have added this part and cited relevant references in the discussion section of the revised manuscript.

Round 2

Reviewer 3 Report

The authors have addressed all of my comments to satisfaction. The manuscript has been improved significantly.